# Workplace Violence Experienced by Personal Care Workers in a District in Seoul, Republic of Korea: A Comparison Study with Office and Service Workers

**DOI:** 10.3390/healthcare12030320

**Published:** 2024-01-26

**Authors:** Mi-Suk Cho, Kyoung-Bok Min, Jin-Young Min

**Affiliations:** 1Veterans Health Service Medical Center, Veterans Medical Research Institute, Seoul 05368, Republic of Korea; misuk9720@naver.com; 2Department of Preventive Medicine, College of Medicine, Seoul National University, Seoul 03080, Republic of Korea; minkb@snu.ac.kr

**Keywords:** healthcare workers, gender-based violence, physical violence

## Abstract

This study compared the level of workplace violence experienced by personal healthcare workers in a district in Seoul, Republic of Korea, with those experienced by workers in office or service jobs. We matched 150 personal care workers with 150 office workers and 150 service workers using a propensity score. Workplace violence was categorized into psychological violence and physical violence. Of the surveyed personal care workers, 53.3% reported experiencing psychological violence, and 42.0% reported experiencing physical violence. After adjusting for self-reported work-related symptoms, personal care workers had significantly higher odds of experiencing psychological violence than office workers (OR = 5.01; 95% CI: 2.80–8.97) or service workers (OR = 7.54; 95% CI: 3.93–14.47). The adjusted odds for physical violence were also significantly higher for personal care workers compared with those for office workers (OR = 5.83; 95% CI: 2.96–11.50) and service workers (OR = 6.00; 95% CI: 2.88–12.49). In terms of specific types of workplace violence, personal care workers were 7–10 times more likely to experience unwanted sexual attention, sexual harassment, and physical violence than office or service workers. We found that personal care workers were more prone to workplace violence than office or service workers, with gender-based or physical violence being the most common types. Considering the negative impact of workplace violence on workers’ well-being and health services, policy updates and interventions focusing on personal care workers are needed to reduce workplace violence, safeguard workers’ rights, and establish a secure working environment.

## 1. Introduction

As life expectancy and the population of older adults increase, there is a growing demand for long-term care in facilities and home-based care for older adults [1]. The recognition that caregiving cannot be addressed solely by older adults themselves, or within their homes, has led to the emergence of a significant national policy [1,2]. In response to demand, the Republic of Korea has implemented several programs to expand the scope of social caregiving. These include the Social Service Voucher Program introduced in May 2007, Long-Term Care (LTC) Insurance for the Elderly introduced in July 2008, the Personal Assistance System for the Disabled introduced in 2011, and the Customized Care Service for the Elderly program introduced in 2020 [2,3].

Personal care workers form the backbone of solutions for caring for older adults. These professionals, trained in implementing the LTC Insurance for the Elderly program, assist older adults who struggle with daily activities due to age or geriatric illnesses [1,4]. Personal care workers are assigned to homes or facilities where older adults live, fulfilling their duty of caring for these adults and protecting those with mobility issues who receive LTC Insurance [1,4]. When delivering services to patients who struggle with communication or physical control or those who are unlikely to improve, these workers endure physical strain. Prior research has shown that personal care workers have reported significant physical issues, such as musculoskeletal pains in the shoulders, hands, wrists, back, and neck, as well as psychological problems, such as depression and sleep problems [5,6,7,8]. Studies on the mental health of care workers in the Republic of Korea have reported that the rate of depression experienced by care workers was 26.2% [9], with 47.2% of depression cases being above mild and 23.3% of anxiety cases being above mild [10]. Care workers who were exposed to violence more than once in the past year showed 2.3 times higher depression rates than those who did not [11]. In addition, the burden placed on care workers was associated with high prevalence rates of poor general health, depression, and more frequent sleep problems [12,13].

Personal care workers are especially susceptible to workplace violence [14,15,16,17]. Workplace violence is a recognized hazard in the healthcare industry. It is defined as any act or threat of physical violence, harassment, intimidation, or other threatening or disruptive behavior that occurs at the work site. Personal care workers experienced more sexual harassment and sexual violence compared with those in other professions, primarily because their main role often involves direct interaction with older adults [14,15,16,17]. Research has shown that these workers frequently face assault from the individuals they care for, particularly when assisting with daily activities, a fundamental aspect of their job [16,17,18,19]. Another study revealed that 40.9% of care workers have been sexually harassed or assaulted by either a service user or their family [20]. Approximately 95% of care workers employed in nursing homes have experienced verbal assault, and 80% have experienced physical assault [21,22].

Numerous studies have examined care workers’ experiences of sexual harassment, physical assault, and verbal abuse [16,17,18,19,20,21,22]. Violence experienced by care workers can negatively affect their physical and psychological well-being and job-related variables (i.e., job satisfaction and burnout). These negative aspects may lead to a lower quality of patient care and indicate the absence of appropriate laws and legislation dealing with safe working environments [23]. However, there is currently no research comparing the extent of physical assault experienced by personal care workers to that experienced by workers in other fields. Risk comparisons of violence experienced by workers in different jobs may be helpful in identifying specific types of risks, their probability, and their severity and in determining the magnitudes of these risks.

In this study, we sought to determine whether personal care workers are at a higher risk of workplace violence compared with workers in other sectors. To accomplish this, we surveyed personal care workers about their experiences with workplace violence and compared their responses with those of office and service sector workers who participated in a nationally representative survey. We used data from the sixth Korean Working Conditions Survey (KWCS) as our nationally representative data. The KWCS is a cross-sectional national survey which monitors comprehensive working conditions and the safety and health of the Korean working population. The KWCS collected a lot of information on workers’ occupations, and it is challenging to distinguish jobs accurately. The KWCS data were used as secondary data to designate the comparison group of personal care workers. We then compared the levels of psychological and physical violence experienced by personal care workers with those experienced by office workers and those experienced by service workers.

## 2. Materials and Methods

### 2.1. Data Source and the Study Population

Two data sources, namely primary and secondary data, were used in this cross-sectional study. For primary data collection, a questionnaire survey was directly administered to personal care workers from June to October 2023. The sample size was calculated using G*Power software, version 3.1.9.2. According to Cohen’s criteria, an effect size of 0.15 is considered a medium effect in gerontology [24]. The significance level was 0.05, and the power was set at 0.9. This calculation estimated that 147 participants were needed for the study. To account for a potential dropout rate of approximately 10%, 160 personal care workers were enrolled. The criteria for inclusion were as follows: age higher than 50 years, active worker for at least 6 months before the study, and consent to participate in the study. Informed consent was obtained from all individuals before they participated in the study. The participants were asked to answer a three-part questionnaire, which included items pertaining to socio-demographic variables, self-reported work-related symptoms, and experiences of workplace violence. The questionnaire items were based on the sixth KWCS conducted in 2020–2021 (Appendix A). Of the 160 personal care workers, 10 who did not answer all the questionnaire items were excluded, resulting in a final study population of 150 personal care workers.

Secondary data were evaluated to compare the likelihood of workplace violence among personal care workers and workers in other sectors, such as office and service workers. This analysis used large-scale national survey data from the sixth KWCS. The Korea Occupational Safety and Health Research Institute conducts the KWCS, a cross-sectional national survey that gathers data on social and occupational health indicators in the work environment. The KWCS was aimed at obtaining information on workers’ occupations, such as industry, employment status (e.g., self-employed or employee), and employment type (regular or temporary), but was limited to the determination of whether the exact job was that of a personal healthcare worker. Out of the 50,538 workers who participated in the KWCS, 14,046 older employees who were aged 50 years or higher, were engaged in office or service work, and for whom complete data for the outcome variables and covariates were available were included. Of these, 7063 and 6983 were office and service workers, respectively.

### 2.2. Workplace Violence

The following questions were posed to assess workplace violence: “Over the last month, have you been subjected to (1) humiliating behavior, (2) threats, (3) unwanted sexual attention, or (4) verbal abuse during your work?” and “Over the past 12 months, have you been subjected to (1) bullying/harassment, (2) sexual harassment, or (3) physical violence during your work?” The participants answered “Yes” or “No”.

Workplace violence was classified into two categories: psychological and physical. Psychological violence encompasses behaviors such as humiliation, threats, unwanted sexual attention, and verbal abuse. Physical violence includes bullying or harassment, sexual harassment, and acts of physical aggression.

### 2.3. Other Variables

An anonymous, self-administered survey was conducted using a three-part questionnaire focusing on socio-demographic variables, self-reported work-related symptoms, and instances of workplace violence. The socio-demographic variables included age (grouped in five-year increments), sex (male or female), education level (below middle school, high-school graduate, or college or higher), and monthly income. Self-reported work-related symptoms were identified as health issues that workers had experienced in the previous 12 months. Participants who reported experiencing pain in the lower back, upper limbs, or lower limbs in the past 12 months were classified as having musculoskeletal pain. Personal care workers were also asked if they had experienced eye pain or headaches, anxiety/depression, and fatigue in the past 12 months, and those who had experienced any of these were classified as having these conditions.

### 2.4. Statistical Analysis

One-to-one matching was performed to evaluate the likelihood of personal care workers experiencing violence in the workplace. This was achieved by comparing personal care workers (who participated in the primary data collection) with office and service workers (who participated in the KWCS), based on propensity scores. These scores were calculated using a logistic regression model, with the status of each employee (whether they were a personal care, office, or service worker) as the dependent variables and four covariates (age, sex, education, and income) as the predictor variables. Personal care workers were matched with office and service workers based on the similarity of their propensity scores, by using the nearest-neighbor matching method without replacement.

Characteristics were compared between personal care and office workers as well as between personal care and service workers. The chi-square test was used to determine statistical differences in these characteristics. To estimate the likelihood of personal care workers experiencing workplace violence, multiple logistic regression analyses were conducted on samples matched by propensity score, by using office and service workers as separate control groups. The logistic models were adjusted for participants’ characteristics and the presence of self-reported work-related symptoms, and the adjusted odds ratio (OR) and 95% confidence intervals (95% CIs) were determined. The significance level was set at <0.05 for all statistical analyses by using a two-tailed test. All statistical analyses were performed using SAS.

## 3. Results

### 3.1. Characteristics of the Participants

Table 1 shows the characteristics of the participants after matching for age, sex, education level, and monthly income by using propensity scores. The study included a total of 450 participants, with 150 each of personal care, office, and service workers. No differences were found in the matched variables between personal care and office workers or between personal care and service workers. Two-thirds of the workers were aged 65 years or higher, and women were more prevalent than men. Over half of the participants had a high school education, and more than 70% earned less than KRW 2,000,000 monthly. Regarding unmatched samples (Appendix A), personal care workers were likelier to be older, female, less educated, and have lower monthly incomes than office and service workers.

### 3.2. Prevalence of Self-Reported Work-Related Symptoms among Workers

Table 2 presents a comparison of the prevalence of self-reported work-related symptoms among personal care, office, and service workers. Comparison of personal care and office workers revealed significantly higher incidence rates of self-reported work-related musculoskeletal symptoms (68.7% vs. 25.3%, *p* < 0.0001), anxiety/depression (20.7% vs. 8.7%, *p* = 0.0033), and fatigue (57.3% vs. 33.3%, *p* < 0.0001) in the former group. However, comparison of personal care and service workers showed significantly lower incidence rates of eye pain/headache (22.7% vs. 84.7%, *p* = 0.0354), anxiety/depression (20.7% vs. 86.7%, *p* = 0.0017), and fatigue (57.3% vs. 92.0%, *p* < 0.0001) in the former group.

### 3.3. Frequency of Psychological and Physical Violence Experienced by Workers

Figure 1 shows the frequency of psychological and physical violence experienced by workers in their workplace. Over half of the personal care workers (53.3%) reported experiencing some form of psychological violence in the past month, a significantly higher percentage than office workers (17.3%, *p* < 0.0001) and service workers (14.7%, *p* < 0.0001). The types of psychological violence experienced by personal care workers included humiliating behaviors (24.7%), threats (16.0%), unwanted sexual attention (25.3%), and verbal abuse (45.3%). The frequency of each type of psychological violence experienced by personal care workers was significantly higher than that experienced by office or service workers (all *p*-values < 0.05).

Just as with psychological abuse, personal care workers reported significantly more instances of physical violence (42.0%) compared with office workers (9.3%) and service workers (8.7%), with the prevalence among personal care workers being nearly five times greater. When examining specific forms of physical violence, personal care workers also reported higher incidences of physical violence (14.7%) and sexual harassment (36.0%) than those reported by office workers (physical violence = 4.7% and sexual harassment = 5.3%) and service workers (physical violence = 1.3% and sexual harassment = 5.3%). However, there was no significant difference in the reported experiences of bullying or harassment among personal care, office, and service workers.

### 3.4. Frequency of Psychological and Physical Violence Experienced by Workers

Table 3 presents the likelihood of personal care workers experiencing psychological or physical violence, with office and service workers serving as reference groups. Personal care workers reported significantly more instances of psychological violence compared with office workers. The adjusted OR (95% CI) for psychological violence was 5.01 (2.80–8.97). The specific ORs (95% CI) for various types of psychological violence were as follows: humiliating behavior, 3.19 (1.52–6.69); threats, 2.87 (1.17–7.03); unwanted sexual attention, 8.13 (3.07–21.50); and verbal abuse, 3.97 (2.20–11.50). Furthermore, the odds of personal care workers experiencing physical violence (adjusted OR = 5.83; 95% CI, 2.96–11.50) were significantly higher than those for office workers. Regarding specific physical violence, the adjusted ORs (95% CI) were 7.93 (3.49–18.03) and 2.71 (1.04–7.08) for sexual harassment and physical violence, respectively.

Comparisons of the likelihood of experiencing psychological or physical violence between personal care and service workers revealed that the results mirrored those of comparisons between personal care and office workers. Furthermore, the OR (95% CI) for psychological violence was significantly higher in personal care workers than in service workers. The adjusted ORs for specific instances of psychological violence were 4.72 (2.09–10.67), 3.63 (1.40–9.38), 9.91 (3.39–28.96), and 6.88 (3.47–13.64) for humiliating behavior, threats, unwanted sexual attention, and verbal abuse, respectively. Except for bullying or harassment, personal care workers have significantly higher odds of experiencing physical violence (adjusted OR = 6.00; 95% CI, 2.88–12.49), sexual harassment (adjusted OR = 7.49; 95% CI, 3.20–17.54), and physical violence (adjusted OR = 8.92; 95% CI, 1.92–41.32) than service workers. Because the participants were predominantly female workers, we performed a sensitivity analysis to test whether our results were affected by sex (Appendix A). We observed consistent results from the sensitivity analysis after excluding male workers.

## 4. Discussion

Our analysis of survey and national data revealed that 53.3% and 42.0%, respectively, of personal care workers encountered at least one form of workplace violence in the past year. These prevalence rates are higher than those reported in earlier studies [25,26,27,28,29]. Our study also provides specific estimates of the prevalence of various forms of abuse experienced by personal care workers: verbal abuse, 45.3%; sexual harassment, 36.0%; unwanted sexual attention, 25.3%; humiliating behavior, 24.7%; and threats, 16.0%. These figures are significantly higher than those reported by office or service workers. When we compared propensity score-matched samples, we found that personal care workers were much more likely to report workplace violence than office or service workers. The increase in the odds of reporting sex-based violence (i.e., unwanted sexual attention and sexual harassment) and physical violence was particularly noticeable.

Workplace violence against personal care workers results in physical and psychological damage at individual levels and has a considerable impact on the delivery of health care services (i.e., leads to poor quality of care delivered, increased absenteeism, and health workers’ decision to leave the field) [23]. Therefore, dealing with workplace violence must be integrated into the functions of organizations rather than being made the responsibility of individual workers. Furthermore, training and education programs should be implemented to improve awareness of the risks of workplace violence and to support good communication skills and coping strategies to deal with violence.

Our findings align with those of previous research indicating a high prevalence of workplace violence among personal care workers. Phaul et al. conducted a survey to assess occupational safety and health experiences, such as injury, biological and chemical exposures, and violence, among 1249 healthcare aides (634 agency-employed and 615 client-employed aides) [25]. They found that approximately 7% and 20% of these individuals reported experiencing physical violence and verbal violence, respectively. Verbal violence was more commonly reported by agency-hired aides than client-hired aides (23% vs. 14%, respectively, *p* < 0.001) [25]. Similarly, Byon et al. conducted an anonymous survey among direct care workers in one large and one medium-sized homecare agency in Chicago. They found that 2.5% and 7.9% of workers reported experiencing violence and threats of violence, respectively, from clients at least once a year [26]. Hanson et al. focused on the prevalence of workplace violence among female homecare workers and found that 61.3% had experienced at least one type of workplace violence in the past year [27], with verbal aggression being the most common (51.5%), followed by sexual harassment (27.6%) and workplace aggression (27.5%). These experiences of violence were associated with increased stress, depression, burnout, and sleep problems. Karlsson et al. reported that 22% of home healthcare aides had experienced at least one instance of verbal abuse in the 12 months preceding their survey [28]. Aides exposed to verbal abuse were 11 times more likely to experience physical abuse. A subsequent study found that aides asked to perform extra tasks beyond their job duties were more likely to experience verbal and physical/sexual abuse [29].

While previous reports have already highlighted a high percentage of personal care workers experiencing workplace violence, our data provide more objective insights by comparing their experiences with those of other workers with similar demographic characteristics. These data also allow us to evaluate the extent to which personal care workers encounter various forms of workplace violence compared with their counterparts.

Although there is no exact mechanism to explain the high odds of workplace violence experiences by personal care workers, understanding their work environment and conditions may help address the issue. These workers primarily provide care for older individuals with moderate to severe health conditions, including hearing impairments and age-related diseases. These patients often communicate their discomfort aggressively, such as through shouting [5]. Unlike hospital care, personal care workers’ main role necessitates close physical contact with patients to assist with activities of daily living [18]. For instance, personal care workers, who are predominantly female, face sexual violence risks while bathing male patients [16,20,30]. Working in confined homes of patients increases the likelihood of verbal abuse [5,15,16,30]. It can be inferred that home-based care workers experience more sexual harassment than facility workers, owing to the confined spaces where they provide their services, often with only the patient’s family members present [19,20]. The worker is in the patient’s personal living space. In the Republic of Korea, 75.8% of home-visiting nurses reported experiencing assault at work, and 67.2% reported workplace assault in the past year. The types of assault included verbal abuse (53.5%), sexual violence (30.3%), threats or harassment (28%), discrimination (30.9%), and physical abuse (2.2%) [31]. This result implies that the work is performed in the patient’s private home; workers who provide services tend to rely more on their resources than the organization’s protection [32,33]. Moreover, it is believed that they are often vulnerable to unsafe events because they work alone and are more likely to be exposed to physical and psychological violence due to the nature of face-to-face work and telecommuting [32,33]. In summary, the challenging work environment of personal care workers, characterized by frequent physical contact and the confined spaces of their clients’ homes, makes them more prone to workplace violence.

This study has several limitations. First, it was a cross-sectional study, and we could not establish causality from the observed relationships. However, our survey recorded long-term incidents of psychological violence over the past 12 months, indicating that the work environment did not change abruptly. Second, we examined two types of data (survey data and national data from the KWCS). To assess the likelihood of workplace violence, we matched personal care workers from the survey with office and service workers who participated in the KWCS. The survey data were collected using the same questionnaire as the KWCS, although the time was different. However, given the recent report on increased workplace violence experienced by healthcare workers because of the COVID-19 pandemic [34], we cannot confirm whether office or service workers matched from the national data were appropriately compared with personal care workers. Third, the surveyed personal care workers were a convenience sample from a district in Seoul, Republic of Korea. They cannot be extrapolated to a broader population, nor can we generalize the research findings. Fourth, the data are based on self-reporting, which may lack objectivity. Workers’ responses to violence may vary, owing to internal and external factors. However, self-reporting is likely to contribute to the relationship in a non-differential way and so may not change the direction of the observed relationship in this study. Lastly, a statistical model was needed to adjust the study’s variables adequately. For example, lifestyle variables (e.g., coffee consumption at work and use of alcohol to relax after work) and occupational variables (e.g., years of work, work dissatisfaction, and frequency of patient handling) are known to be beneficial or risk factors for workplace violence [35]. In addition, the frequency, nature, significance, reporting, and perpetration of workplace violence could be essential information that affected the results.

## 5. Conclusions

Our findings show higher odds of experiencing workplace violence among personal care workers than among office or service workers. Considering the negative impact of workplace violence on workers’ well-being and health services, policy updates and interventions focusing on personal care workers are needed to reduce workplace violence, safeguard workers’ rights, and establish a secure working environment.

## Figures and Tables

**Figure 1 healthcare-12-00320-f001:**
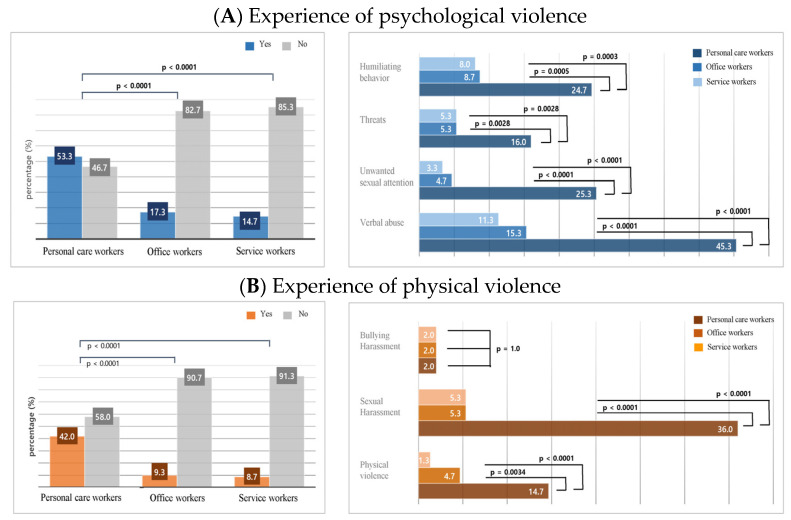
Comparisons of workers by the experience of psychological and physical violence at work.

**Table 1 healthcare-12-00320-t001:** Participants’ characteristics after propensity score matching.

Characteristics	Personal Care Workers (*n* = 150)		KWCS					
			Office Workers (*n* = 150)			Service Workers (*n* = 150)		
	*n*	(%)	*n*	(%)	*p*-Value ^a^	*n*	(%)	*p*-Value ^b^
Age, years								
~54	7	(4.67)	7	(4.67)	0.9986	10	(6.67)	0.9326
55–59	12	(8.00)	12	(8.00)		13	(8.67)	
60–64	36	(24.00)	38	(25.33)		37	(24.67)	
65–69	59	(39.33)	58	(38.67)		58	(38.67)	
70–74	32	(21.33)	30	(20.00)		30	(20.00)	
≥75	4	(2.67)	5	(3.33)		2	(1.33)	
Sex						150		
Male	9	(6.00)	10	(6.67)	0.8126	10	(6.67)	0.8126
Female	141	(94.00)	140	(93.33)		140	(93.33)	
Education level								
Below middle school	39	(26.00)	38	(25.33)	0.9905	41	(27.33)	0.8352
High school graduate	81	(54.00)	82	(54.67)		83	(55.33)	
College or more	30	(20.00)	30	(20.00)		26	(17.33)	
Monthly income (KRW)								
Less than 2,000,000	109	(72.67)	109	(72.67)	1	108	(72.00)	0.9399
2,000,000~3,000,000	32	(21.33)	32	(21.33)		34	(22.67)	
More than 3,000,000	9	(6.00)	9	(6.00)		8	(5.33)	

KWCS: Korean Working Conditions Survey; ^a^
*p*-values between personal care workers and office workers were determined using the chi-square test. ^b^ *p*-values between personal care workers and service workers were determined using the chi-square test.

**Table 2 healthcare-12-00320-t002:** Comparisons of self-reported work-related symptoms between personal care workers, office workers, and service workers.

Self-Reported Work-Related Symptoms	Personal Care Workers (*n* = 150)		KWCS					
			Office Workers (*n* = 150)			Service Workers (*n* = 150)		
	*n*	(%)	*n*	(%)	*p*-Value ^a^	*n*	(%)	*p*-Value ^b^
Musculoskeletal symptoms								
Yes	103	(68.7)	38	(25.3)	<0.0001	104	(69.3)	0.7925
No	47	(31.3)	112	(74.7)		46	(31.7)	
Eye/headache								
Yes	34	(22.7)	21	(14.0)	0.0524	127	(84.7)	0.0354
No	116	(77.3)	129	(86.0)		23	(15.3)	
Anxiety/Depression								
Yes	31	(20.7)	13	(8.7)	0.0033	130	(86.7)	0.0017
No	119	(79.3)	137	(91.3)		20	(23.3)	
Fatigue								
Yes	86	(57.3)	50	(33.3)	<0.0001	138	(92.0)	<0.0001
No	64	(42.7)	100	(66.7)		12	(18.0)	

KWCS: Korean Working Conditions Survey ^a^
*p*-values between personal care workers and office workers were determined using the chi-square test. ^b^
*p*-values between personal care workers and service workers were determined using the chi-square test.

**Table 3 healthcare-12-00320-t003:** Odds for psychological or physical violence at work for personal care workers.

Types of Violence	Unadjusted Model			Adjusted Model *		
	OR	(95% CI)	*p*-Value	OR	(95% CI)	*p*-Value
Personal care workers vs. Office workers as reference						
Psychological violence	5.45	(3.21–9.27)	<0.0001	5.01	(2.80–8.97)	<0.0001
Humiliating behavior	3.48	(1.77–6.87)	0.0003	3.19	(1.52–6.69)	0.0021
Threats	3.38	(1.47–7.79)	0.0043	2.87	(1.17–7.03)	0.0215
Unwanted sexual attention	6.93	(2.98–16.11)	<0.0001	8.13	(3.07–21.50)	<0.0001
Verbal abuse	4.58	(2.65–7.92)	<0.0001	3.97	(2.20–7.17)	<0.0001
Physical violence	7.03	(3.72–13.32)	<0.0001	5.83	(2.96–11.50)	<0.0001
Bullying/harassment	1.00	(0.20–5.04)	1.0000	0.82	(0.14–4.84)	0.8252
Sexual harassment	9.98	(4.55–21.92)	<.0001	7.93	(3.49–18.03)	<0.0001
Physical violence	3.51	(1.45–8.49)	0.0053	2.71	(1.04–7.08)	0.0414
Personal care workers vs. Service workers as reference						
Psychological violence	7.65	(3.82–11.58)	<0.0001	7.54	(3.93–14.47)	<0.0001
Humiliating behavior	4.80	(1.89–7.63)	0.0002	4.72	(2.09–10.67)	0.0002
Threats	3.83	(1.47–7.79)	0.0043	3.63	(1.40–9.38)	0.0079
Unwanted sexual attention	9.94	(3.75–25.81)	<0.0001	9.91	(3.39–28.96)	<0.0001
Verbal abuse	6.94	(3.57–11.80)	<0.0001	6.88	(3.47–13.64)	<0.0001
Physical violence	7.63	(3.97–14.69)	<0.0001	6.00	(2.88–12.49)	<0.0001
Bullying/harassment	1.00	(0.20–5.04)	1.0000	0.82	(0.11–6.03)	0.8420
Sexual harassment	9.98	(4.55–21.92)	<0.0001	7.49	(3.20–17.54)	<0.0001
Physical violence	12.72	(2.93–55.14)	0.0007	8.92	(1.92–41.32)	0.0052

* Adjusted model was adjusted by participants’ characteristics and self-reported work-related symptoms.

## Data Availability

The datasets analyzed during the current study are available from the corresponding author upon reasonable request.

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
