# Peer review of "Workplace Violence Experienced by Personal Care Workers in a District in Seoul, Republic of Korea: A Comparison Study with Office and Service Workers"

_healthcare, 2024, doi:10.3390/healthcare12030320_

Round 1
Reviewer 1 Report
Comments and Suggestions for Authors
Dear authors, I read with interest your work and found the study well conducted and properly described. A few suggestion are needed to fully evaluate your work.
1) the introduction is lacking the knowledge gap. Even though in lines 45-53 you correctly weigh the problem with magnitude data, you forgot to underline why is it important a direct comparison with other professionals. for instance, has ever been conducted a specific preventive programme or do you believe that your results will contribute in designing one? what else?
2) line 71: did you mean 147 participant per arm?
3)line 73, typo: must have been
4) lines 77-78: i think it is necessary to report the questionnaire in supplementary materials.
5)lines 114-133: is it possible that the chosen personal healthcare workers were also already included in the KWCS service workers sample? I think that the reader need assurance that they were not.
6)lines 264-269: at this point We were expecting a reason for your finding so different from the literature (you declare it at line 232)
7) Can you give us a hint of why these results are important? what can we do to limit or eradicate this phenomenon? if not properly addressed, what are the consequences of so stressed personal healthcare workers?
Kind Regards,
Author Response
Responses to Reviewer 1
We are grateful for this comprehensive review. We have revised our manuscript following your suggestions and comments. Our point-by-point responses to your comments are summarized below in black font.
I read with interest your work and found the study well conducted and properly described. A few suggestion are needed to fully evaluate your work.
1) the introduction is lacking the knowledge gap. Even though in lines 45-53 you correctly weigh the problem with magnitude data, you forgot to underline why is it important a direct comparison with other professionals. for instance, has ever been conducted a specific preventive programme or do you believe that your results will contribute in designing one? what else?
[Response] By your comment, the Introduction section has been more described as follows;
“Numerous studies have examined the experiences of care workers with sexual harassment, physical assault, and verbal abuse [16-22]. Violence experience of care workers can negatively affect their physical and psychological well-being and job-related variables (i.e., job satisfaction and burnout). These negative aspects may lead to lower quality of patient care and indicate the absence of appropriate laws and legislation dealing with safe working environments. [23]. However, there is currently no research comparing the extent of physical assault experienced by direct care workers to that experienced by workers in other fields. Risk comparison of violence experienced by workers in different jobs may be helpful to identify specific types of risk, their probability, and their severity and determine the magnitudes of the risks.”
2) line 71: did you mean 147 participant per arm?
[Response] Yes, the participants means 160 personal care workers. For the clear description, we have revised the sentence as follows;
“160 participants must be over 50 years old.”
3) line 73, typo: must have been
[Response] The typo has been revised correctly.
4) lines 77-78: i think it is necessary to report the questionnaire in supplementary materials.
[Response] By your comment, we have reported the questionnaire in supplementary materials.
"The questionnaire items were based on the 6th KWCS conducted during 2020-2021 (Table S1)."
5) lines 114-133: is it possible that the chosen personal healthcare workers were also already included in the KWCS service workers sample? I think that the reader need assurance that they were not.
[Response] By your comment, we have included the limitation of the KWCS in terms of identifying personal healthcare workers as follows;
"The KWCS investigated information on workers’ occupations – for example, industry, employment status (self-employed, employee, etc.), and employment type (regular, temporary) – but it was limited to determine whether the exact job is a personal healthcare worker. "
6) lines 264-269: at this point We were expecting a reason for your finding so different from the literature (you declare it at line 232)
[Response] Reference has been revised correctly with some changes of the sentence as follows;
"Unlike hospital care, personal care workers' main role necessitates close physical contact with patients to assist with daily activities of daily living [18]. "
- Byon, H.D.; Storr, C.L.; Lipscomb, J. Latent classes of caregiver relationships with patients: workplace violence implications. Geriatr Nurs (Minneap). 2017, 38, 291‐295.
7) Can you give us a hint of why these results are important? what can we do to limit or eradicate this phenomenon? if not properly addressed, what are the consequences of so stressed personal healthcare workers?
[Response] By your comment, we have added the following descriptions in the Discussion and Conclusion sections.
"Workplace violence against personal care workers affects physical and psychological damage at individual levels and has a significant impact on the delivery of health care services (i.e., poor quality of care delivered, increased absenteeism, and health workers' decision to leave the field) [23]. Therefore, dealing with workplace violence must be integrated into the organization rather than being made the responsibility of individual workers. At the same time, training and education programs should be incorporated to improve awareness of the risks of workplace violence and to support good communication skills and coping strategies to deal with violence."
“In conclusion, our findings show higher odds of experiencing workplace violence among personal care workers than among office or service workers. Considering the negative impact of workplace violence on workers' well-being and health services, policy updates and interventions focusing on individual care workers are needed to reduce workplace violence, safeguard workers' rights, and establish a secure working environment.”
We hope our approach is acceptable to you.
Reviewer 2 Report
Comments and Suggestions for Authors
To the esteemed authors,
The paper is excellently written, with only a little suggestion to further improve it:
1. The abstract should discuss the implications of the findings rather than just reporting the results.
2. The discussion should provide more detailed explanation for why personal healthcare workers are more prone to workplace violence, specifically referring to the data reported in lines 283 - 286.
3. Please eliminate the restriction of the study to Conclusion.
4.The conclusion should include a comprehensive explanation of how the current findings contribute to the advancement of existing knowledge.
5. What strategies may be implemented to mitigate violence in the workplace, safeguard workers' rights, and establish a secure working environment? Please provide comprehensive insights from your perspective in the debate. By doing so, individuals can assess the eloquence of your ideas.
Author Response
Responses to Reviewer 2
We are grateful for this comprehensive review. We have revised our manuscript following your suggestions and comments. Our point-by-point responses to your comments are summarized below in black font.
The paper is excellently written, with only a little suggestion to further improve it:
1) The abstract should discuss the implications of the findings rather than just reporting the results.
[Response] By your comment, we have included the implications of the findings in the abstract as follows;
"Considering the negative impact of workplace violence on workers’ well-being and health services, policy updates and interventions focusing on individual care workers are needed to reduce workplace violence, safeguard workers' rights, and establish a secure working environment."
2) The discussion should provide more detailed explanation for why personal healthcare workers are more prone to workplace violence, specifically referring to the data reported in lines 283 - 286.
[Response] By your comment, we have added more detailed explanation for why personal healthcare workers are more prone to workplace violence as follows;
"This result implies that the work is performed in the patient's private home; workers who provide services tend to rely more on their resources than the organization's protection [32,33]. Moreover, it is believed that they are often vulnerable to unsafe events because they work alone and are more likely to be exposed to physical and psychological violence due to the nature of face-to-face and telecommuting [32,33]. "
3) Please eliminate the restriction of the study to Conclusion.
4) The conclusion should include a comprehensive explanation of how the current findings contribute to the advancement of existing knowledge.
[Response] By your comment, we have revised the Conclusion as follows;
“In conclusion, our findings show higher odds of experiencing workplace violence among personal care workers than among office or service workers. Considering the negative impact of workplace violence on workers' well-being and health services, policy updates and interventions focusing on individual care workers are needed to reduce workplace violence, safeguard workers' rights, and establish a secure working environment.”
5) What strategies may be implemented to mitigate violence in the workplace, safeguard workers' rights, and establish a secure working environment? Please provide comprehensive insights from your perspective in the debate. By doing so, individuals can assess the eloquence of your ideas.
[Response] By your comment, the following description has been added in the Discussion section;
"Workplace violence against personal care workers affects physical and psychological damage at individual levels and has a significant impact on the delivery of health care services (i.e., poor quality of care delivered, increased absenteeism, and health workers' decision to leave the field) [23]. Therefore, dealing with workplace violence must be integrated into the organization rather than being made the responsibility of individual workers. At the same time, training and education programs should be incorporated to improve awareness of the risks of workplace violence and to support good communication skills and coping strategies to deal with violence."
We hope our approach is acceptable to you.
Reviewer 3 Report
Comments and Suggestions for Authors
This study aimed to investigate WPV among Korean HCPs.
I have the following comments:
1: The title and the abstract did not mention where the study was conducted. It should be made clear that this study came from one district in South Korea.
2: The questions used for determining different types of WPV should be mentioned in the methods section.
3: The authors did not mention when the data were collected. Previous reports showed increased WPV during the COVID-19 pandemic.
4: Did HCPs and KWCs fill out the same questionnaire at the same time?
5: Why did the authors apply an effect size of 0.25? This could have falsely reduced the least acceptable sample size.
6: Table 1 is not needed. The authors have built their analysis and conclusions based on the matched sample. So, the authors should move it to the supplementary section.
7: The depression rates are very high, especially in service workers. Are there any explanations? I also recommend highlighting the fact that medical conditions were self-reported not clinically determined. The authors may use the term (self-reported work-related symptoms) rather than (work-related diseases) across the manuscript.
8: Table 4 should show the results of the unadjusted ORs (95% CIs) as well.
9: Several studies investigating WPV among HCWs were ignored.
10: The limitations section should describe the lifestyle and occupational variables that were not collected and might have affected the results.
11: The limitations section should mention that essential information related to WPV, such as the WPV frequency, nature, significance, reporting, and perpetration was not mentioned.
12: Lines 270-275: I suggest rephrasing.
13: The authors should consider adding a paragraph suggesting solutions.
14: The conclusion section should recommend further research avoiding the limitations of the current study.
15: It seems that the authors, in their first draft, used text citations as names rather than numbers. However, they did not remove the names in some paragraphs and overused the names in others. This should be revised across the entire study.
Comments on the Quality of English Language
English language editing is needed.
Author Response
Responses to Reviewer 3
We are grateful for this comprehensive review. We have revised our manuscript following your suggestions and comments. Our point-by-point responses to your comments are summarized below in black font.
This study aimed to investigate WPV among Korean HCPs. I have the following comments:
1) The title and the abstract did not mention where the study was conducted. It should be made clear that this study came from one district in South Korea.
[Response] By your comment, we have revised the title and the abstract as follow;
“Workplace violence of personal care workers in a district of Seoul, South Korea”
“This study compared the level of workplace violence experienced by personal healthcare workers in one district of Seoul, South Korea with those in office or service jobs.”
2) The questions used for determining different types of WPV should be mentioned in the methods section.
[Response] All questionnaires used in the current study used the same questions as the KCWS, which is a large-scale national survey to investigate occupational and environmental risk factors, and provide preliminary data for improving working conditions.
By your comment, the questions to determine different types of WVP included in the Method section as follows;
"The following questions assessed workplace violence; "Over the last month, during the course of your work have you subjected to any of 1) humiliating behavior, 2) threats, 3) unwanted sexual attention, or 4) verbal abuse?" and "Over the past 12 months, during the course of your work have you subjected to 1) bullying/harassment, 2) sexual harassment, or 3) physical violence?" Participants answered "Yes" or "No." "
3) The authors did not mention when the data were collected. Previous reports showed increased WPV during the COVID-19 pandemic.
[Response] We conducted a questionnaire survey directly with personal care workers for our primary data collection from June to October 2023. The survey was done the period which the COVID-19 pandemic was in a slight lull.
“We conducted a questionnaire survey directly with personal care workers for our primary data collection from June to October 2023".
4) Did HCPs and KWCs fill out the same questionnaire at the same time?
[Response] We conducted a questionnaire survey directly with personal care workers for our primary data collection from June to October 2023. Limitation due to variation in the timing of two surveys has been revised as follows;
"Second, we examined two types of data (survey data and national data from the KWCS). To assess the likelihood of workplace violence, we matched personal care workers from the survey with office and service workers who participated in the KWCS. The survey data was collected using the same questionnaire as the KWCS, even though the time was different. However, given the recent report on increased work-place violence related to COVID 19 pandemic in healthcare employees [34], we cannot confirm whether the sample (office or service worker matched from the national data) is appropriately compared with personal care workers. "
5) Why did the authors apply an effect size of 0.25? This could have falsely reduced the least acceptable sample size.
[Response] Using G*Power software, we calculated the sample size. According to Cohen’s criteria, an effect size of 0.15 was considered a medium effect in gerontology [24]. The significance level was 0.05, and the power was set at 0.9. This calculation estimated that 147 participants were needed for our study. To account for a potential dropout rate of approximately 10%, we enrolled 160 personal care workers.
6) Table 1 is not needed. The authors have built their analysis and conclusions based on the matched sample. So, the authors should move it to the supplementary section.
[Response] By your comment, we have moved Table 1 to Supplementary material as Table S2.
7) The depression rates are very high, especially in service workers. Are there any explanations? I also recommend highlighting the fact that medical conditions were self-reported not clinically determined. The authors may use the term (self-reported work-related symptoms) rather than (work-related diseases) across the manuscript.
[Response] As you commented, our data showed that service workers reported very high rates of anxiety/depression. To understand why service workers are vulnerable to anxiety/depression, additional studies are needed to identify risk factors at individual and societal levels, particularly in older workers over 50 years old. Thus, we think it is impossible to explain the observed phenomenon.
By your comment, we have revised the term (self-reported work-related symptoms) throughout the manuscript.
8) Table 4 should show the results of the unadjusted ORs (95% CIs) as well.
[Response] By your comment, we have included the unadjusted ORs (95% CIs) in Table 3.
9) Several studies investigating WPV among HCWs were ignored.
[Response] To study the high odds of workplace violence of personal care workers, all authors faithfully reviewed the previous studies.
The classification of health workers maps occupation categories into five broad groupings: health professionals, health associate professionals, personal care workers in health services, health management and support personnel, and other health service providers not elsewhere classified. In the current study, we focused on workplace violence of personal care workers in health services, rather than overall health workers. All authors believe that we faithfully reviewed the previous studies.
In the Introduction section,
"Personal care workers are especially susceptible to workplace violence [14-17]. Workplace violence is a recognized hazard in the healthcare industry. It is any act or threat of physical violence, harassment, intimidation, or other threatening disruptive behavior that occurs at the work site. Personal care workers had more experience of sexual harassment and sexual violence compared with those in other professions, primarily because their main job role often involves direct interaction with older adults [14-17]. Research has shown that these workers frequently face assault from the individuals they care for, particularly when assisting with daily activities, a fundamental aspect of their job [16-19]. Another study revealed that 40.9% of care workers have been sexually harassed or assaulted by either the service user or their family [20]. Approximately 95% of care workers employed in nursing homes have experienced verbal assault, and 80% have experienced physical assault [21,22]."
In the Discussion section,
"Our findings align with those of previous research indicating a high prevalence of workplace violence among personal care workers. Phaul et al. (2010) conducted a sur-vey assessing occupational safety and health experiences, such as injury, biological and chemical exposures, and violence, among 1249 healthcare aides (634 agency-employed, 615 client-employed) [25] They found that roughly 7% reported physical violence, and 20% experienced verbal violence. Verbal violence was more commonly reported by agency-hired aides than client-hired aides (23% vs. 14% respectively, p < 0.001) [25]. Similarly, Byon et al. (2016) carried out an anonymous survey of direct care workers in one large and one medium-sized home care agency in Chicago, finding that 2.5% and 7.9% of workers reported experiencing violence and threats of violence, respectively, from clients at least once a year [26]. Hanson et al. (2015) focused on the prevalence of workplace violence among female homecare workers, finding that 61.3% had experienced at least one type of workplace violence in the past year [27]. Verbal aggression was the most common (51.5%), followed by sexual harassment (27.6%) and workplace aggression (27.5%). These experiences of violence were associated with increased stress, depression, burnout, and sleep problems. Karlsson et al. (2019) reported that 22% of home healthcare aides had experienced at least one instance of verbal abuse in the 12 months preceding the survey [28]. Aides exposed to verbal abuse were 11 times more likely to experience physical abuse. A subsequent study found that aides asked to per-form extra tasks beyond their job duties were more likely to experience verbal and physical/sexual abuse [29]."
10) The limitations section should describe the lifestyle and occupational variables that were not collected and might have affected the results.
11) The limitations section should mention that essential information related to WPV, such as the WPV frequency, nature, significance, reporting, and perpetration was not mentioned.
[Response] To your comment, we have include the additional limitation of this study as follows;
"Lastly, the statistical model was needed to adjust the study's variables adequately. For example, lifestyles (i.e., coffee consumption at work and use of alcohol to relax after work) and occupational variables (i.e., years of work, work dissatisfaction, and frequency of patient handling) are known to be beneficial or risk factors of workplace violence [35]. In addition, the frequency, nature, significance, reporting, and perpetration of workplace violence could be essential information that affected the results."
12) Lines 270-275: I suggest rephrasing.
[Response] By your comment, we have revised the sentence as follows;
“Although there is no exact mechanism to explain the high odds of workplace violence against personal care workers, understanding their work environment and conditions may help address it.”
13) The authors should consider adding a paragraph suggesting solutions.
[Response] By your comment, we have included the following sentences in the Discussion section as follows;
"Workplace violence against personal care workers affects physical and psychological damage at individual levels and has a significant impact on the delivery of health care services (i.e., poor quality of care delivered, increased absenteeism, and health workers' decision to leave the field) [23]. Therefore, dealing with workplace violence must be integrated into the organization rather than being made the responsibility of individual workers. At the same time, training and education programs should be incorporated to improve awareness of the risks of workplace violence and to support good communication skills and coping strategies to deal with violence."
14) The conclusion section should recommend further research avoiding the limitations of the current study.
[Response] By your comment, we have revised the Conclusion as follows;
“In conclusion, our findings show higher odds of experiencing workplace violence among personal care workers than among office or service workers. Considering the negative impact of workplace violence on workers' well-being and health services, policy updates and interventions focusing on individual care workers are needed to reduce workplace violence, safeguard workers' rights, and establish a secure working environment.”
15) It seems that the authors, in their first draft, used text citations as names rather than numbers. However, they did not remove the names in some paragraphs and overused the names in others. This should be revised across the entire study.
[Response] By your comment, we have removed the names in some paragraphs and overused the names in others across the entire study.
We hope our approach is acceptable to you.
Reviewer 4 Report
Comments and Suggestions for Authors
I have read the "Workplace violence of personal healthcare workers: a comparison study with office and service workers". Overall, it is a very well-written study that is scientifically sound and very clear as to what it assessed, and how it sits with the literature, acknowledging the limitations of a cross-sectional study.
See below my comments:
-The authors discuss the objectives of their analysis of secondary data in lines 82-90 and they present the type of primary data they collected in the paragraphs before and after the section, without clearly stating the objective of their primary data (in the same way they did about their secondary data). It would have been more beneficial if, at the end of the introduction, they stated the two types of objectives and proceeded in their methodology discussing about their two types of data in greater detail.
-As the study included Korean samples, I would expect the authors to cite some other findings relevant to the subject in the Korean context drawing for example, from the type of secondary data they worked on or other sources.
Author Response
Responses to Reviewer 4
We are grateful for this comprehensive review. We have revised our manuscript following your suggestions and comments. Our point-by-point responses to your comments are summarized below in black font.
I have read the "Workplace violence of personal healthcare workers: a comparison study with office and service workers". Overall, it is a very well-written study that is scientifically sound and very clear as to what it assessed, and how it sits with the literature, acknowledging the limitations of a cross-sectional study. See below my comments:
1) The authors discuss the objectives of their analysis of secondary data in lines 82-90 and they present the type of primary data they collected in the paragraphs before and after the section, without clearly stating the objective of their primary data (in the same way they did about their secondary data). It would have been more beneficial if, at the end of the introduction, they stated the two types of objectives and proceeded in their methodology discussing about their two types of data in greater detail.
[Response] By your comment, we have revised the Introduction as follows;
"In this study, we sought to determine whether personal care workers are at a higher risk of workplace violence compared with workers in other sectors. To accomplish this, we surveyed personal care workers about their experiences with workplace violence and compared their responses with those of office and service sector workers who participated in a nationally representative sample data. We used the 6th Korean Working Conditions Survey (KWCS) data as nationally representative data. The KWCS is a cross-sectional national survey to monitor comprehensive working conditions and the safety and health of the Korean working population. The KWCS investigated a lot of information on workers' occupations, and it is challenging to distinguish jobs accurately. The KWCS was used as secondary data to designate the comparison group of the personal care workers. We then compared the experiences of psychological and physical violence between personal care workers and office workers, and be-tween personal care workers and service workers."
2) As the study included Korean samples, I would expect the authors to cite some other findings relevant to the subject in the Korean context drawing for example, from the type of secondary data they worked on or other sources.
[Response] By your comment, we have added the reference relevant to the subject in the Korean context. The following references were included in the revised manuscript.
"Studies on the mental health of care workers in South Korea have reported that the rate of depression experienced by care workers was 26.2% [9], 47.2% of depression above mild, and 23.3% of anxiety above mild [10]. Care workers who were exposed to violence more than once in the past year showed 2.3 times higher depression than those who did not [11]."
"In South Korea, 75.8% of home-visiting nurses reported experiencing assault at work, and 67.2% reported workplace assault in the past year. The types of assault included verbal abuse (53.5%), sexual violence (30.3%), threats or harassment (28%), discrimination (30.9%), and physical abuse (2.2%) [31]. This result implies that the work is performed in the patient's private home; workers who provide services tend to rely more on their resources than the organization's protection [32,33]. Moreover, it is believed that they are often vulnerable to unsafe events because they work alone and are more likely to be exposed to physical and psychological violence due to the nature of face-to-face and telecommuting [32,33]. "
References
- Hwang, B.R.; Yoo, E.K. A study on the risk factors of the client violence experiences of care workers working in nursing homes. Journal of Social Science. 2015, 31, 1-28.
- Lee, J. M.; Hong, M.H.; Jang, K.W. Convergence study about the relationship among emotional dissonance, depression and anxiety in care service workers -Focused on the moderating effects of emotional intelligence. Journal of the Korea Convergence Society, 2021, 12, 341-351.
- Jeon, G.S.; You, S.J.; Kim, M.G.; Kim, Y.M. Correlates of depressive symptoms and stress among Korean women careworkers for older adults dwelling in community. Korean J Occup Health Nurs, 2017, 26, 10-18.
- Kim, E.; Choi, H.; Yoon, J.Y.; Who cares for visiting nurses? Workplace violence against home visiting nurses from public health centers in Korea. Int J Environ Res Public Health. 2020, 17, 4222.
- Park, E.; Kim, J.H. The Experiences of workplace violence toward nurses in hospitals in jeju province,South Korea. Korean J Occup Health Nurs, 2011, 20, 212–220.
- Lee, I.S.; Lee, K.O.; Kang, H.S.; Park, Y.H. Violent experiences and coping among home visiting health care workers in Korea. J Korean Acad Nurs, 2012, 42, 66–75.
We hope our approach is acceptable to you.
Reviewer 5 Report
Comments and Suggestions for Authors
In the manuscript, the authors reported on the effects of job type on workplace violence in the workplace. The manuscript is well written, and provides detailed descriptions of the analyses and results. Personal healthcare professionals are a group that have not been adequately studied in previous studies. The findings have theoretical and practical implications, and contributes to the literature of healthcare. Please see specific comments below:
1. In the second paragraph, it would be helpful to introduce a bit more research on mental health among personal healthcare workers, e.g., specific symptoms of depression, and negative outcomes of sleep disorders.
2. The operational definition of workplace violence should be provided earlier in the introduction section.
3. Please provide references for the selection of the small sample size in the the power analysis using G*Power.
4. In the sample, the participants were predominantly females. Only 6% were male workers, would it be helpful to exclude those and ran a sensitivity analysis to validate the results? This analysis may be more suitable when considering subtypes of physical violence.
5. Although the study focuses on experiences of psychological and physical violence, were there any psychological assessments of negative outcomes of violence, e.g.., depression, anxiety, etc.?
6. Based on the findings of the study, possible interventions are expected to be discussed.
Author Response
Responses to Reviewer 5
We are grateful for this comprehensive review. We have revised our manuscript following your suggestions and comments. Our point-by-point responses to your comments are summarized below in black font.
In the manuscript, the authors reported on the effects of job type on workplace violence in the workplace. The manuscript is well written, and provides detailed descriptions of the analyses and results. Personal healthcare professionals are a group that have not been adequately studied in previous studies. The findings have theoretical and practical implications, and contributes to the literature of healthcare. Please see specific comments below:
1) In the second paragraph, it would be helpful to introduce a bit more research on mental health among personal healthcare workers, e.g., specific symptoms of depression, and negative outcomes of sleep disorders.
[Response] By your comment, we have introduced more research on mental health as follows;
"Studies on the mental health of care workers in South Korea have reported that the rate of depression experienced by care workers was 26.2% [9], 47.2% of depression above mild, and 23.3% of anxiety above mild [10]. Care workers who were exposed to violence more than once in the past year showed 2.3 times higher depression than those who did not [11]. As well, the burden of care workers was associated with the high prevalence of poor general health, depression, and more frequent sleep problems [12,13]."
2) The operational definition of workplace violence should be provided earlier in the introduction section.
[Response] By your comment, we have added the definition of workplace violence in the Introduction section.
"Workplace violence is a recognized hazard in the healthcare industry. It is any act or threat of physical violence, harassment, intimidation, or other threatening disruptive behavior that occurs at the work site."
3) Please provide references for the selection of the small sample size in the power analysis using G*Power.
[Response] Using G*Power software, we calculated the sample size. According to Cohen’s criteria, an effect size of 0.15 was considered a medium effect in gerontology [24]. The signifi-cance level was 0.05, and the power was set at 0.9. This calculation estimated that 147 participants were needed for our study. To account for a potential dropout rate of approximately 10%, we enrolled 160 personal care workers.
- Brydges, C.R. Effect size guidelines, sample size calculations, and statistical power in gerontology. Innov Aging. 2019, 3, igz036.
4) In the sample, the participants were predominantly females. Only 6% were male workers, would it be helpful to exclude those and ran a sensitivity analysis to validate the results? This analysis may be more suitable when considering subtypes of physical violence.
[Response] By your comment, we have performed the sensitivity analysis by excluding male workers of only 6%. The results have been in Table S3 as follows;
5) Although the study focuses on experiences of psychological and physical violence, were there any psychological assessments of negative outcomes of violence, e.g.., depression, anxiety, etc.?
[Response] We agree on the importance of whether there are any psychological assessments of adverse outcomes of violence. Unfortunately, we did not collect data to assess it.
6) Based on the findings of the study, possible interventions are expected to be discussed.
[Response] By your comment, we have added the following descriptions in the Discussion and Conclusion sections.
"Workplace violence against personal care workers affects physical and psychological damage at individual levels and has a significant impact on the delivery of health care services (i.e., poor quality of care delivered, increased absenteeism, and health workers' decision to leave the field) [23]. Therefore, dealing with workplace violence must be integrated into the organization rather than being made the responsibility of individual workers. At the same time, training and education programs should be incorporated to improve awareness of the risks of workplace violence and to support good communication skills and coping strategies to deal with violence."
“In conclusion, our findings show higher odds of experiencing workplace violence among personal care workers than among office or service workers. Considering the negative impact of workplace violence on workers' well-being and health services, policy updates and interventions focusing on individual care workers are needed to reduce workplace violence, safeguard workers' rights, and establish a secure working environment.”
We hope our approach is acceptable to you.
Round 2
Reviewer 3 Report
Comments and Suggestions for Authors
I have no more comments.